# Complications of Non-Alcoholic Fatty Liver Disease in Extrahepatic Organs

**DOI:** 10.3390/diagnostics10110912

**Published:** 2020-11-07

**Authors:** Wataru Tomeno, Kento Imajo, Takuya Takayanagi, Yu Ebisawa, Kosuke Seita, Tsuneyuki Takimoto, Kanami Honda, Takashi Kobayashi, Asako Nogami, Takayuki Kato, Yasushi Honda, Takaomi Kessoku, Yuji Ogawa, Hiroyuki Kirikoshi, Yasunari Sakamoto, Masato Yoneda, Satoru Saito, Atsushi Nakajima

**Affiliations:** 1Department of Gastroenterology, International University of Health and Welfare Atami Hospital, 13-1 Higashikaigancho, Atami-shi, Shizuoka 413-0012, Japan; tomeno-ykh@umin.ac.jp (W.T.); t_takayanagi1987@yahoo.co.jp (T.T.); you.ebisawa603@gmail.com (Y.E.); innocent_society@yahoo.co.jp (K.S.); ttakimoto8@gmail.com (T.T.); kanamin.ni@gmail.com (K.H.); takaomi0027@gmail.com (T.K.); yasakamo@iuhw.ac.jp (Y.S.); 2Department of Gastroenterology and Hepatology, Yokohama City University Graduate School of Medicine, 3-9 Fukuura, Kanazawa-ku, Yokohama 236-0004, Japan; kento318@yokohama-cu.ac.jp (K.I.); tkhkcb@gmail.com (T.K.); a06m071@yahoo.co.jp (A.N.); y-honda@umin.ac.jp (Y.H.); t.kato222@iuhw.ac.jp (T.K.); ogaway@yokohama-cu.ac.jp (Y.O.); yoneda-ycu@umin.ac.jp (M.Y.); ssai1423@yokohama-cu.ac.jp (S.S.); 3Department of Clinical Laboratory, Yokohama City University Hospital, 3-9 Fukuura, Kanazawa-ku, Yokohama 236-0004, Japan; hkirikos@yokohama-cu.ac.jp

**Keywords:** non-alcoholic fatty liver disease, non-alcoholic steatohepatitis, extrahepatic complications, chronic kidney disease, colorectal cancer, major depressive disorder, gastroesophageal reflux disease, obstructive sleep apnea syndrome, periodontitis, hypothyroidism

## Abstract

Non-alcoholic fatty liver disease (NAFLD) is now recognized as the most common chronic liver disease worldwide, along with the concurrent epidemics of metabolic syndrome and obesity. Patients with NAFLD have increased risks of end-stage liver disease, hepatocellular carcinoma, and liver-related mortality. However, the largest cause of death among patients with NAFLD is cardiovascular disease followed by extrahepatic malignancies, whereas liver-related mortality is only the third cause of death. Extrahepatic complications of NAFLD include chronic kidney disease, extrahepatic malignancies (such as colorectal cancer), psychological dysfunction, gastroesophageal reflux disease, obstructive sleep apnea syndrome, periodontitis, hypothyroidism, growth hormone deficiency, and polycystic ovarian syndrome. The objective of this narrative review was to summarize recent evidences about extrahepatic complications of NAFLD, with focus on the prevalent/incident risk of such diseases in patients with NAFLD. To date, an appropriate screening method for extrahepatic complications has not yet been determined. Collaborative care with respective experts seems to be necessary for patient management because extrahepatic complications can occur across multiple organs. Further studies are needed to reveal risk profiles at baseline and to determine an appropriate screening method for extrahepatic diseases.

## 1. Introduction

Over the last few decades, the incidences of non-alcoholic fatty liver disease (NAFLD) and non-alcoholic steatohepatitis (NASH) have been increasing worldwide, along with the concurrent epidemics of metabolic syndrome and obesity. A recent meta-analysis estimated that the overall global prevalence of NAFLD is very high at 25.2% [1]. In this analysis, the regional prevalence of NAFLD was reported to be highest in the Middle East (31.7%) and South Africa (30.4%). Thus, NAFLD has been recognized as the most common chronic liver disease worldwide. Compared with matched, controlled populations without NAFLD, patients with NAFLD have increased risks of end-stage liver disease, hepatocellular carcinoma (HCC) [2], and liver-related mortality [1]. Therefore, both hepatologists and patients must recognize hepatic conditions, especially those characterized by liver fibrosis [3,4]. However, hepatic involvement is just one component of the multiorgan manifestation of NAFLD. In fact, a recent multicenter cohort study revealed that the largest cause of death among patients with NAFLD is cardiovascular disease (CVD) followed by extrahepatic malignancies, whereas liver-related mortality is only the third cause of death [3]. NAFLD represents a liver component of metabolic syndrome and is associated with risk factors for metabolic syndrome, including obesity [5], diabetes mellitus [6], and dyslipidemia [7]. Recently, a growing body of evidence has been collected that supports the notion that NAFLD should be treated as an early mediator of systemic diseases and metabolic syndrome, as well as liver-specific diseases [8,9]. The objective of this narrative review was to summarize recent evidences about extrahepatic complications of NAFLD, with focus on the prevalent/incident risk of such diseases in patients with NAFLD. The association between NAFLD and CVD is not mentioned in this article because it is discussed in great detail in another article in this special issue.

## 2. Chronic Kidney Disease

Chronic kidney disease (CKD) is a worldwide public health problem, with possible adverse outcomes that include end-stage renal disease (ESRD), CVD, and premature death. CKD is defined by a sustained reduction in the glomerular filtration rate or evidence of structural or functional abnormalities of the kidneys on urinalysis, biopsy, or imaging [10]. The two major risk factors of CKD are hypertension and diabetes mellitus, which are also major risk factors for NAFLD. A recent meta-analysis demonstrated that patients with NAFLD had a significantly higher risk of incident CKD than those without NAFLD (random-effects hazard ratio (HR), 1.37; 95% confidence interval (CI), 1.20-1.53) [11]. In other studies, patients with NAFLD were similarly reported to have a higher prevalence of CKD than patients without NAFLD (Table 1) [12,13,14,15,16,17,18,19]. Importantly, the majority of these studies demonstrated that NAFLD was independently associated with CKD even after adjustments for risk factors including age, sex, body mass index (BMI), hypertension, diabetes mellitus, smoking, and hyperlipidemia. The possible pathogeneses linking NAFLD and CKD include the upregulation of the renin–angiotensin system and the impairment of antioxidant defense. An excess dietary fructose may also contribute to NAFLD and CKD. Moreover, CKD may mutually aggravate NAFLD and associated metabolic disturbances through altered intestinal barrier function and microbiota composition, alterations in glucocorticoid metabolism, and the accumulation of uremic toxic metabolites [20].

## 3. Extrahepatic Malignancies

### 3.1. Colorectal Cancer

Colorectal cancer is the third most commonly diagnosed malignancy and the fourth leading cause of cancer-related deaths in the world, accounting for about 1.4 million new cases and almost 700,000 deaths in 2012 [21]. A recent meta-analysis of observational studies suggested that NAFLD was independently associated with a moderately increased prevalence and incidence of colorectal adenomas and cancer (Odds ratio (OR), 1.28 for prevalent adenomas and 1.56 for prevalent cancer; HR, 1.42 for incident adenomas and 3.08 for cancer) [22]. In addition, several studies have reported the prevalent risk of colorectal cancer in patients with NAFLD (Table 2) [23,24,25,26,27,28,29]. However, these retrospective studies were mainly reported from Asia. Therefore, future studies are needed to confirm the true risks of colorectal cancer among various NAFLD populations.

The presence of metabolic syndrome, especially diabetes mellitus and obesity, is a well-known risk factor for colorectal cancer [30,31]. However, whether NAFLD is associated with an increased risk of colorectal cancer simply as a consequence of the shared metabolic risk factors, or whether NAFLD itself may contribute to the development of colorectal cancer, is uncertain [22]. Regarding the former possibility, hyperinsulinemia induced by insulin resistance promotes carcinogenesis by stimulating the proliferation pathway through its effect on insulin receptors on cancer cells. In addition, hyperinsulinemia increases the expression of insulin-like growth factor (IGF)-1, which has mitogenic and anti-apoptotic activities that are more potent than those of insulin and can act as a stimulus for the growth of preneoplastic and neoplastic cells [30]. Regarding the latter possibility, patients with NAFLD reportedly have reduced serum levels of adiponectin, which has anti-carcinogenic effects. This mechanism is due to the ability of adiponectin to stop the growth of colon cancer cells through AMPc-activated protein kinase (AMPK) and to induce a caspase-dependent pathway resulting in endothelial cell apoptosis [32].

### 3.2. Other Malignancies

Although the association with colorectal cancer and NAFLD is most frequently reported, several studies have reported an increased risk of extrahepatic malignancies in other organs in patients with NAFLD. A recent study from Korea reported that female patients with NAFLD had an increased risk of developing breast cancer (HR, 1.92; 95% CI, 1.15–3.20) [33]. Furthermore, a few studies have suggested increased risks of developing gastric cancer [34,35], pancreatic cancer [35,36], prostate cancer [37], and esophageal cancer [33,35] in patients with NAFLD.

## 4. Psychological Dysfunction

Major depressive disorder (MDD) is an important public health problem. Among those affected, 28% experience a moderate degree of functional impairment, whereas 59% experience severe reductions in their normal functional ability [38]. MDD is often comorbid with other chronic diseases, such as cardiovascular disease, arthritis, asthma, and diabetes mellitus, and the presence of MDD reportedly worsens the health outcomes of these associated conditions [39]. Similarly, an association between chronic liver disease and MDD has also been demonstrated. Patients with NAFLD reportedly have a prevalence of MDD of 27.2%, which is higher than that of the general population [40]. Our previous study showed that NAFLD patients with comorbid MDD have severe histological steatosis and a higher NAFLD activity score, as well as significantly higher levels of serum aminotransferase, γ-glutamyl transpeptidase, and ferritin than age- and sex-matched NAFLD patients without MDD. They also demonstrated that NAFLD patients with MDD had a poor response to the standard care for NAFLD, consisting mainly of lifestyle modifications [41]. The pathogenesis linking NAFLD and MDD remains uncertain. However, a recent study investigated possible changes in brain tissue volumes in patients with NAFLD. They reported that the brain volumes of white and gray matter were significantly reduced in patients with NAFLD, compared with control subjects. Accordingly, this reduction in brain volume might be related to a higher risk of depression in patients with NAFLD [42]. In addition, NAFLD reportedly correlated with other types of psychological dysfunction, such as cognitive impairment and Alzheimer’s disease. Cerebrovascular alterations, neuroinflammation, and brain insulin resistance are thought to be key factors in the pathogenesis of these diseases [43].

## 5. Gastroesophageal Reflux Disease

Gastroesophageal reflux disease (GERD) and NAFLD are commonly associated with metabolic syndrome, especially visceral obesity. Visceral obesity increases the intragastric pressure because of the accumulation of adipose tissue in the whole abdominal cavity, which may induce an abnormal gastroesophageal reflux leading to GERD symptoms [44]. Patients with NAFLD reportedly have a high prevalence of symptoms of GERD [45,46]. Several studies have shown that GERD is associated with sleep problems [47]. Symptoms of GERD (such as heartburn or regurgitation) during the night are thought to cause multiple short periods of awakening, leading to sleep fragmentation. Taketani et al. reported that nearly 30% of Japanese patients with biopsy-proven NAFLD had insomnia, which was independently associated with GERD symptoms. They also demonstrated that treatment with a proton pump inhibitor could relieve both insomnia and the GERD symptoms [48]. Changes in the secretion of hormones (such as cortisol, leptin, and ghrelin) and increased insulin resistance because of a short sleep duration are thought to increase the risks of obesity and diabetes [49,50]. Therefore, GERD and insomnia may lead to the progression and worsening of NAFLD.

## 6. Obstructive Sleep Apnea Syndrome

Patients with obstructive sleep apnea syndrome (OSAS) have repetitive episodes of shallow or paused breathing during sleep that cause a reduction in blood oxygen saturation. This chronic intermittent hypoxia (CIH) induces increased oxidative stress, the generation of reactive oxygen species (ROS), and the release of inflammatory cytokines, resulting in systemic inflammation that drives the exacerbation of NAFLD and the progression to liver fibrosis [51,52]. Musso et al. conducted a meta-analysis of eighteen cross-sectional studies and reported that OSAS is associated with an increased risk (independent of age, sex, and BMI) of NAFLD (OR, 2.99), NASH (OR, 2.37), and advanced fibrosis (OR, 2.30) [53]. Some non-randomized observational studies have reported a beneficial effect of continuous positive airway pressure (CPAP) on surrogate markers of NAFLD [53,54]. However, the currently available evidence from randomized controlled trials [55,56] does not suggest that the treatment of OSAS with CPAP can reverse the exacerbation of NAFLD.

## 7. Periodontitis

Individuals with chronic periodontitis reportedly have a significantly increased risk of developing CVD, including atherosclerosis, myocardial infarction, and stroke [57]. Furthermore, a two-way relationship exists between diabetes and periodontitis—diabetes increases the risk of periodontitis, while periodontal inflammation negatively affects glycemic control [58]. Similarly, an association between periodontitis and NAFLD has also been reported in various studies [59]. Yoneda et al. investigated the detection frequency of *Porphyromonas gingivalis*, which is well known as a major causative agent of periodontitis, in 150 biopsy-proven NAFLD patients and 60 healthy controls [60]. They found that the frequencies of *P. gingivalis* detection in patients with biopsy-proven NAFLD (46.7%) and NASH (52.0%) were significantly higher than that in non-NAFLD control subjects (21.7%). Moreover, they also demonstrated that non-surgical periodontal treatments in NAFLD patients ameliorated the serum levels of Aspartate aminotransferase (AST) and Alanine aminotransferase (ALT). Although further large-scale clinical trials are needed, periodontal treatments may be useful supportive measures in the management of patients with NAFLD.

## 8. Endocrinopathies

### 8.1. Hypothyroidism

Pagadala et al. reported that the prevalence of hypothyroidism was significantly higher among patients with biopsy-proven NAFLD than among age-, sex-, race-, and BMI-matched control subjects (21% vs. 9.5%) [61]. Moreover, they found that NASH was associated with hypothyroidism, in a manner that was independent of diabetes mellitus, dyslipidemia, hypertension, and age. In addition, Chung et al. investigated the prevalence of NAFLD in 2324 cases with hypothyroidism, compared with age- and sex-matched controls [62]. They found that the prevalence of NAFLD was significantly higher not only in patients with overt hypothyroidism, but also in patients with subclinical hypothyroidism (even in the upper normal range of thyroid-stimulating hormone (TSH) levels), than in subjects with euthyroidism, independent of known metabolic risk factors [62].

### 8.2. Adult Growth Hormone Deficiency

Growth hormone (GH) profoundly reduces visceral fat, which plays an important role in the development of NAFLD. Moreover, GH directly reduces lipogenesis in hepatocytes [63]. Adult growth hormone deficiency (AGHD) is characterized by metabolic abnormalities associated with visceral obesity, impaired quality of life, and increased mortality. Nishizawa et al. reported that the prevalence of NAFLD in patients with AGHD was significantly higher than in age-, sex-, and BMI-matched healthy controls (77% vs. 12%, *p* < 0.001) [64]. They also demonstrated the effectiveness of GH replacement therapy for NAFLD, although the sample size was too small to assess the true effectiveness.

### 8.3. Polycystic Ovarian Syndrome

Polycystic ovarian syndrome (PCOS), also known as hyperandrogenic anovulation, is one of the most common endocrine disorders in women of reproductive age. Several studies have demonstrated that the prevalence of NAFLD was significantly higher among patients with PCOS, and that hyperandrogenism was an independent risk factor for NAFLD [65,66]. Although PCOS and NAFLD share similar metabolic comorbidities such as obesity, diabetes mellitus, dyslipidemia, and metabolic syndrome, androgen excess itself is thought to increase insulin resistance [67]. The prevalence of PCOS in patients with NAFLD is still unclear. However, a cross-sectional study from Australia reported that the prevalence of PCOS was very high at 71% (10/14) among female NAFLD patients of reproductive age (20–45 years) [68].

## 9. Discussion

Although considerable evidence and suggestions have been collected to reveal associations between NAFLD and extrahepatic complications, most of these reports were based on cross-sectional or observational studies with short follow-up periods. An appropriate screening method for extrahepatic complications has not yet been clearly described in any guidelines or guidances, such as the European guidelines from the European Association for the Study of the Liver (EASL) [69], guidance from the American Association for the Study of Liver Disease (AASLD) [70], or guidelines from the Japanese Society of Gastroenterology (JSG) [71]. However, intensive surveillance and early intervention for these extrahepatic complications might benefit the patients with NAFLD. For extrahepatic malignancies, the NAFLD patients who have never undergone colonoscopy should undergo total colonoscopy at least once. Further studies are needed to reveal the risk profiles at baseline and to determine appropriate screening methods for extrahepatic diseases. There are some limitations in this narrative review. First, we did not conduct a statistical analysis about bias risk in our selection of referenced literatures. Second, the association between NAFLD and CVD was not mentioned in this article because it was discussed in great detail in another article in this special issue. Recently, clinical trials examining new therapeutic drugs for NAFLD/NASH that act via various mechanisms are being performed in several countries [72]. The influence of these drugs on the risks and outcomes of extrahepatic complications should be considered in these clinical trials. To date, the influence of life style modifications to extrahepatic complications are unclear, therefore it might be the candidate for future studies. Since extrahepatic complications occur across multiple organ systems, collaborative care with respective experts is needed for the management of patients with NAFLD. In conclusion, compared to healthy controls, patients with NAFLD have higher prevalence/incidence risk of extrahepatic complications in multiple organs. Not only hepatologists, but also patients with NAFLD, should be aware of these increased risks for extrahepatic diseases.

## Figures and Tables

**Table 1 diagnostics-10-00912-t001:** Studies examining the risk of developing chronic kidney disease (CKD) among patients with non-alcoholic fatty liver disease (NAFLD).

Author, Year of Publication	Study Population	Country	Diagnosis of NAFLD	Results
*Retrospective studies*				
Yun et al., 2009 [12]	37,085 healthy subjects who underwent health examinations	Korea	Serum ALT > 40 IU/L (patients with hepatitis B or C, or alcoholic liver disease were excluded)	ALT > 40 IU/L group had a significantly higher creatinine level than ALT < 40 IU/L group (0.9 mg/dL vs. 0.8 mg/dL).
Targher et al., 2010 [13]	80 patients with biopy-proven NASH and 80 control subjects matched for age, sex, and BMI	Italy	Histological	NASH was independently associated with an increased prevalence of CKD (25% vs. 3.7%). Adjusted OR, 6.14 (95% CI, 1.60–12.8).
Yasui et al., 2011 [14]	174 patients with biopsy-proven NAFLD	Japan	Histological	Prevalence of CKD was significantly higher among NASH group (21%) than among simple steatosis group (6%). Adjusted OR, 2.46 (*p* = 0.11; not significant because of dependency on hypertension)
Sirota et al., 2012 [15]	11,469 volunteers who participated in the National Health and Nutrition Examination Survey (NHANES 1988–1994)	United States	Ultrasonography (other chronic liver disease were excluded)	NAFLD was not associated with the prevalence of CKD after adjustment for components of metabolic syndrome.
Sinn et al., 2017 [16]	41,430 adult men and women without CKD at baseline who underwent repeated health check-up examinations	Korea	Ultrasonography	NAFLD was associated with an increased risk of CKD development. Adjusted HR, 1.22 (95% CI, 1.04–1.43). The risk of CKD increased progressively with increases in the NAFLD fibrosis score.
Park et al., 2019 [17]	262,619 newly diagnosed patients with NAFLD and 769,878 propensity score (1:3)-matched non-NAFLD patients from the Truven Health MarketScan Database (2006–2015)	United States	Ultrasonography	Patients with NAFLD had a 41% increased risk of developing advanced (stages 3–5) CKD compared with non-NAFLD patients. Adjusted HR, 1.41 (95% CI, 1.36–1.46).
*Prospective studies*				
Chang et al., 2008 [18]	8329 healthy male volunteers with normal baseline kidney functions (Follow-up duration: 3.2 years)	Korea	Ultrasonography	NAFLD was independently associated with the development of CKD. Adjusted relative risk, 1.55.
Targher et al., 2008 [19]	2103 patiens with type 2 diabetes mellitus	Italy	Ultrasonography	NAFLD was associated with increased risk of CKD (OR, 1.87) independently of age, sex, BMI, hypertension, diabetes duration, HbA1c, or LDL-cholesterol.

ALT: Alanine aminotransferase, NAFLD: Non-alcoholic fatty liver disease, NASH: Non-alcoholic steatohepatitis, BMI: Body mass index, CKD: chronic kidney disease, OR: Odds ratio, CI: Confidence interval, HR: Hazard ratio, HbA1c: Hemoglobin A1c, LDL: Low-density lipoprotein.

**Table 2 diagnostics-10-00912-t002:** Retrospective studies examining the risk of colorectal adenomas and cancer in patients with NAFLD.

Author, Year of Publication	Study Population	Country	Diagnosis of NAFLD	Results
Wong et al., 2011 [23]	199 patients with NAFLD and 181 healthy controls	China	Magnetic resonance spectroscopy (*n* = 64) and liver biopsy (*n* = 135)	NASH was independently associated with colorectal adenomas (adjusted OR, 4.89; 95% CI, 2.04–11.70) and advanced colorectal neoplasms (adjusted OR, 5.34; 95% CI, 1.92–14.84).
Stadlmayr et al., 2011 [24]	1211 subjects who underwent screening colonoscopy	Austria	Ultrasonography (other chronic liver diseases were excluded)	The prevalence of colorectal lesions was 34% in the NAFLD group and 21.7% in the control group (*p* < 0.001). NASH was independently associated with colorectal adenomas (adjusted OR, 1.47; 95% CI, 1.08–2.00).
Touzin et al., 2011 [25]	94 patients with biopsy-proven NAFLD and 139 patients without NAFLD	United States	Histological	No significant difference in the prevalence of colonic adenomas (24.4% in NAFLD patients compared with 25.1% in non-NAFLD patients) was seen. NAFLD patients had significantly greater numbers of colonic adenomas than non-NAFLD patients (*p* = 0.016).
Lee et al., 2012 [26]	5517 women (831 patients with NAFLD) who underwent health check-up	Korea	Ultrasonography	NAFLD was independently associated with colorectal adenomatous polyps (adjusted OR, 1.94; 95% CI, 1.11–3.40) and colorectal cancer (adjusted OR, 3.08; 95% CI, 1.02–9.34).
Huang et al., 2013 [27]	1522 health-check individuals who underwent two consecutive colonoscopies (no adenomas were detected at first colonoscopy)	Taiwan	Ultrasonography	NAFLD was an independent risk factor (OR, 1.45; 95% CI, 1.07–1.98) for adenoma formation after a negative baseline colonoscopy.
Ahn et al., 2017 [28]	26,540 subjects who underwent a first-time colonoscopy as part of a health check-up program	Korea	Ultrasonography	NAFLD was independently associated with colorectal neoplasia (adjusted OR, 1.10; 95% CI, 1.03–1.17) and advanced colorectal neoplasia (adjusted OR, 1.21; 95% CI, 0.99–1.47).
Blackett et al., 2020 [29]	123 patients with biopsy-proven NAFLD and controls without liver disease matched by age, sex, and endoscopist	United States	Histological	Patients with biopsy-proven NAFLD had a significantly higher colorectal adenoma prevalence independently of hyperlipidemia, diabetes, and obesity (OR, 1.74; 95% CI, 1.05–2.88).

NAFLD: Non-alcoholic fatty liver disease, NASH: Non-alcoholic steatohepatitis, OR: Odds ratio, CI: Confidence interval.

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
