# Peer review of "Complications of Non-Alcoholic Fatty Liver Disease in Extrahepatic Organs"

_diagnostics, 2020, doi:10.3390/diagnostics10110912_

Round 1
Reviewer 1 Report
Request minor corrections for the paper; English corrections and modifications required, reference confirmation and notation
Author Response
Thank you very much for your important comment. Your suggestions have been fully addressed in the revised manuscript, which we feel has been now greatly improved as a result.
Comment 1
English corrections and modifications required, reference confirmation and notation.
Response to comment 1
Thank you very much for your important suggestion. We confirmed English notation and made modifications. We also confirmed the reference.
Reviewer 2 Report
Dear Authors,
I have read with interest the review article “Complications of Nonalcoholic Fatty Liver Disease in Extrahepatic Organs”.
This is a very interesting review article, and you should be congratulated for your work and identification of the included references.
Below are my comments.
The review article will benefit if the intention/objective with the review article, in the Abstract and in the full text article, is described with care. A clear objective helps the reader to understand what is coming next. Otherwise, the context (line 23-32) reads well.
It is not clear the Material and methods used. There should be some sentences on the methodological approach.
- How is the referenced literature collected/searched for? How exhaustive were the searches?
- How is it assessed as reliable (bias risk), and that no other publication is overlooked which may be much better to include?
The sentence “Recently, a growing body of evidence has been collected that supports the notion that NAFLD should be treated as an early mediator of systemic diseases and metabolic syndrome, as well as a liver-specific disease.” needs to be supported with references.
Then: “This review focuses on extrahepatic complications of NAFLD, highlighting recent key studies”. Key studies – how is this determined? If the methods are clear, then the word is easier to understand. Perhaps the author can write ‘studies that …..”
- In the Discussion, the authors write: “An appropriate screening method for extrahepatic complications has not yet been clearly described in any guidelines or guidances, such as the European guidelines from the European Association for the Study of the Liver (EASL) [67], guidance from the American Association for the Study of Liver Disease (AASLD) [68], or guidelines from the Japan Society of Gastroenterology (JSG) [69].” However, I lack to see the conclusion in their own work. What is the message to the reader, having read the article?
- There should be more limitations listed in the Discussion. This will not mean that the current work is without a value. The value would depend on the reliability /trustworthiness and the transparency of the information presented within the article.
- Part of the Discussion seems to contain conclusions, but there must be a sentence or two which should link to the objectives of this overview.
- There is no mention of lifestyle modifications and whether these may impact complications in NAFLD.
Chavdar Pavlov
14.10.2020
Author Response
Thank you very much for your important comment. Your suggestions have been fully addressed in the revised manuscript, which we feel has been now greatly improved as a result.
Comment 1
The review article will benefit if the intention/objective with the review article, in the Abstract and in the full text article, is described with care. A clear objective helps the reader to understand what is coming next. Otherwise, the context (line 23-32) reads well.
Response to comment 1
Thank you for your useful comment. The objective of this narrative review is to summarize recent evidences about extrahepatic complications of NAFLD, with focus on the prevalent/incident risk of such diseases in patients with NAFLD. We revised the abstract and the Introduction section to mention the objective of this article.
Comment 2
It is not clear the Material and methods used. There should be some sentences on the methodological approach.
- How is the referenced literature collected/searched for? How exhaustive were the searches?
- How is it assessed as reliable (bias risk), and that no other publication is overlooked which may be much better to include?
Response to comment 2
Thank you very much for your important indication. This is a narrative review, not a systematic review. Therefore, we did not conducted statistical analysis about bias risk in our selection of studies. The description of the methodological approach is considered to be not necessary for a narrative review.
Comment 3
The sentence “Recently, a growing body of evidence has been collected that supports the notion that NAFLD should be treated as an early mediator of systemic diseases and metabolic syndrome, as well as a liver-specific disease.” needs to be supported with references.
Response to comment 3
Thank you very much for having pointed out the omission of this point. We added references to this sentence.
Comment 4
“This review focuses on extrahepatic complications of NAFLD, highlighting recent key studies”. Key studies – how is this determined? If the methods are clear, then the word is easier to understand. Perhaps the author can write ‘studies that …..”
Response to comment 4
Thank you very much for your useful comment. As mentioned above, we did not conducted statistical analysis in this narrative review. Therefore it may become our subjective opinion. We revised this sentence and deleted the word “key studies”.
Comment 5
In the Discussion, the authors write: “An appropriate screening method for extrahepatic complications has not yet been clearly described in any guidelines or guidances, such as the European guidelines from the European Association for the Study of the Liver (EASL) [67], guidance from the American Association for the Study of Liver Disease (AASLD) [68], or guidelines from the Japan Society of Gastroenterology (JSG) [69].” However, I lack to see the conclusion in their own work. What is the message to the reader, having read the article?
Response to comment 5
Thank you very much for your important suggestion. In this review, we have summarized that patients with NAFLD had high prevalent/incident risk of extrahepatic diseases. We could not detect any evidence about appropriate screening methods for extrahepatic complications. However, intensive surveillance and early intervention for these extrahepatic complications might benefit the patients with NAFLD. We revised the Discussion section to mention this point.
Comment 6
There should be more limitations listed in the Discussion. This will not mean that the current work is without a value. The value would depend on the reliability /trustworthiness and the transparency of the information presented within the article.
Response to comment 6
Thank you very much for your useful comment. We listed the limitations of this review article in the Discussion section.
Comment 7
Part of the Discussion seems to contain conclusions, but there must be a sentence or two which should link to the objectives of this overview.
Response to comment 7
Thank you for your important comment. We revised the Discussion section and added our conclusion.
Comment 8
There is no mention of lifestyle modifications and whether these may impact complications in NAFLD.
Response to comment 8
Thank you very much for your important suggestion. To date, the influence of life style modifications to extrahepatic complications are still unclear. We revised the Discussion section to mention this point.